# Purification and Characterization of Antibodies Directed against the α-Gal Epitope

Andreas Zappe [1],* , Julia Rosenlöcher [1], Guido Kohla [1], Stephan Hinderlich [2] and Maria Kristina Parr [3]

[1]  Department of Molecular Biochemistry, Octapharma Biopharmaceuticals GmbH, 12489 Berlin, Germany; julia.rosenlocher@octapharma.com (J.R.); guido.kohla@octapharma.com (G.K.)

[2]  Department of Life Sciences and Technology, Beuth University of Applied Sciences, 13353 Berlin, Germany; hinderlich@beuth-hochschule.de

[3]  Institute of Pharmacy, Freie Universität Berlin, 14195 Berlin, Germany; maria.parr@fu-berlin.de

*  Correspondence: a.zappe@fu-berlin.de

**Abstract:** The α-Gal epitope is an immunogen trisaccharide structure consisting of *N*-acetylglucosamine (GlcNAc)β1,4-galactose (Gal)α1,3-Gal. It is presented as part of complex-type glycans on glycoproteins or glycolipids on cell surfaces of non-primate mammalians. About 1% of all antibodies in human sera are specific toward α1,3-Gal and are therefore named as anti-α-Gal antibodies. This work comprises the purification and characterization of anti-α-Gal antibodies from human immunoglobulin G (IgG). A synthetically manufactured α Gal epitope affinity resin was used to enrich anti-α-Gal antibodies. Selectivity experiments with purified antibodies were carried out using enzyme-linked immunosorbent assays (ELISA), Western blotting, and erythrocyte agglutination. Furthermore, binding affinities toward α-Gal were determined by surface plasmon resonance (SPR) and the IgG distribution of anti α Gal antibodies (83% IgG2, 14% IgG1, 2% IgG3, 1% IgG4) was calculated applying ELISA and immunodiffusion. A range of isoelectric points from pH 6 to pH 8 was observed in 2D gel electrophoresis. Glycan profiling of anti α Gal antibodies revealed complex biantennary structures with high fucosylation grades (86%). Additionally, low amounts of bisecting GlcNAc (15%) and sialic acids (13%) were detected. The purification of anti-α-Gal antibodies from human IgG was successful, and their use as detection antibodies for α Gal-containing structures was evaluated.

**Keywords:** α-Gal epitope; allergy; anti-α-Gal antibody; IgG subclasses; *N*-glycosylation; surface plasmon resonance; gel electrophoresis

## 1. Introduction

The α-Gal epitope is an immunogen glycan antigen composed of the structure *N*-acetylglucosamine (GlcNAc)β1,4-galactose-(Gal)α1,3-Gal [1,2]. The epitope is located on complex-type glycans of proteins and lipids of non-primate mammals [3]. It was originally discovered by Karl Landsteiner in 1935 [4]. Primates do not express the α-Gal epitope because of an evolutionary inactivation of the α1,3-galactosyltransferase [5,6], except α-Gal epitopes on human tumor cells [7–9]. Up to 1% of all IgG antibodies in human sera show selectivity toward α1,3-Gal on glycoproteins [10–13] and glycolipids [14–16].

The primate immune system is continuously stimulated by antigenic carbohydrate structures on cell surfaces of gastrointestinal bacteria [17,18]. Consequently, primate organisms produce anti-α-Gal antibodies as polyclonal antibodies. Galili et al. conducted pioneer research in terms of purification and selectivity analysis of anti-α-Gal antibodies. They discovered anti-α-Gal producing B-cells in lymphoid tissue along the gastrointestinal tract [19]. The presence of anti-α-Gal antibodies leads to severe immune reactions up to anaphylactic shock when exposed to α1,3-Gal. However, the unconjugated glycan epitope has no effect on the immune system. It is considered a hapten due to its small size and can only develop its immunogenic effect in combination with bigger antigens. Synthetic α-Gal

haptens are currently designed and used in animal models to investigate different types of immune responses [20].

The α-Gal epitope was found in pig tissue [21,22] and thyroglobulin of various species, including bovine [23]. Therefore, it remains challenging to carry out xenotransplantations without α-Gal-mediated rejection reactions [24,25]. The α-Gal epitope may also be present on recombinant proteins produced in murine cell systems, e.g., cetuximab (brand name Erbitux$^{TM}$), a chimeric mouse–human IgG1 therapeutic antibody recombinantly expressed in SP2/0 cells [26].

A high prevalence of hypersensitivity reactions to cetuximab has been reported in some areas of the United States [26–28], correlated with a high titer of cetuximab-specific IgE antibodies [29]. The analysis of these IgE antibodies revealed high selectivity for α-Gal epitopes located on the Fab regions [29,30]. The location of patients with high titers of anti-cetuximab IgE antibodies geographically overlapped with regions of reported "red meat allergy" [31–34]. This allergy could be linked to tick bites by *Amblyomma americanum*, known as Lone Star tick [35–39]. It could be shown that components in tick saliva trigger the immune system and induce immunogenic effects by α-Gal-containing substances [30].

We intended to obtain novel analytical results on the properties of anti-α-Gal IgG antibodies, purified from a commercial, liquid, ready-to-use, intravenous human IgG concentrate (Octagam®). From the pharmaceutical and health care perspective, the α-Gal epitope is a dangerous immunogen epitope, which may be located on recombinant glycoproteins. The availability of purified anti-α-Gal antibodies as a detection tool could prevent safety risks and rejection reactions. Aspects such as binding behavior, selectivity, and sensitivity must be known before specific applications such as rapid and straightforward detection can be analyzed. We consider antibody purification, characterization, and selectivity experiments as the first important steps for α-Gal detection. In addition, it was our concern to use classic methods for the evaluation and characterization of purified antibodies. We chose a human IgG concentrate as a source for the purification of anti-α-Gal antibodies because anti-α-Gal IgG make up a large proportion of human serum and their influence on the immune system is virtually unknown. The analyses carried out here provide a good basis for further experiments dealing with therapeutic applications of anti-α-Gal antibodies. Anti-α-Gal antibodies in patients' blood may lead to a more effective uptake and presentation of antigen from antigen-presenting cells (APCs). Increased activity of the immune system has already been identified on the mouse model using α-Gal-coated influenza virus [40].

## 2. Materials and Methods

If not indicated otherwise, chemicals were purchased from Sigma Aldrich GmbH, Munich, Germany, and Merck KGaA, Darmstadt, Germany. Biotinylated glycans and the α-Gal affinity resin (Galα1,3-Galβ1,4-GlcNAcβ-OCH$_2$CH$_2$CH$_2$NH$_2$) were purchased from GlycoNZ, Auckland, Australia. Octagam® (purity 95%) was provided as a liquid, ready-to-use, intravenous human IgG concentrate (10% $w/v$, 100 mg/mL) by Octapharma Biopharmaceuticals GmbH, Dessau, Germany.

### 2.1. Purification of Anti-α-Gal Antibodies

The purification of anti-α-Gal antibodies from Octagam® IgG concentrate was performed using an ÄktaPure 25 Chromatography System (GE Healthcare, Freiburg, Germany). A glass column (Tricorn 5/50, 5 mm column diameter, volume 1.16 mL, GE Healthcare, Freiburg, Germany) was manually packed with 1 mL of α-Gal affinity resin (GlycoNZ, Auckland, Australia) and equilibrated with PBS (0.5 mL/min). The absorption signal at 214 nm was monitored, and an Octagam® sample (100 mg/mL) was applied. The affinity matrix was loaded with three samples of human IgG concentrate, each with a volume of 5 mL. The chromatography was carried out at room temperature with a constant flow rate of 0.5 mL/min. The third sample injection was followed by an additional washing step with 5 mL of PBS. The elution of retained antibodies was performed with

500 mM galactose in PBS at 0.5 mL/min. A further wash step with 100 mM glycine pH 2.0 followed at the same flow rate. The Gal-eluted antibody fraction was collected separately and stored at 4 °C. The buffer of samples was exchanged to PBS pH 7.4 and concentrated by Amicon® Ultra centrifugal filters (cutoff: 30 kDa, Merck, Darmstadt, Germany). The Protein concentration was determined by the Pierce bicinchoninic acid (BCA) protein assay kit (Thermo Fisher Scientific, Schwerte, Germany) using IgG as a standard, and adjusted to 1 mg/mL.

### 2.2. Conjugation of Anti-α-Gal Antibodies with Horseradish Peroxidase

The conjugation of purified anti-α-Gal antibodies with horseradish peroxidase (HRP) was accomplished using the HRP conjugation kit of Abcam (Cambridge, United Kingdom). The conjugation reaction was carried out according to the manufacturer's instructions. In brief, modifier reagent was mixed with the anti-α-Gal antibody solution (1 mg/mL) in a ratio of 1:10. This solution was added to HRP and incubated for 3 h in the dark at room temperature. A quenching reagent was added to the modifier–antibody solution in a ratio of 1:10. The mixture was again incubated for 30 min in the dark, without further removal of excess HRP.

### 2.3. Characterization of the Purified Anti-α-Gal Antibody

#### 2.3.1. ELISA Binding Assays

Biotinylated glycan epitopes were immobilized on a streptavidin precoated, flat-bottom, 96-well plate (Thermo Fisher Scientific, Schwerte, Germany) by incubation of 1 µg/mL glycan in PBS overnight at 4 °C while shaking (300 rpm). Unspecific binding was blocked with Roti-Block™ blocking buffer (Carl Roth GmbH, Karlsruhe) for 1 h. The well plate was washed between each incubation step 3 times with PBS/0.1% (*v/v*) Tween 20. Purified anti-α-Gal antibodies were applied with the indicated concentrations. Samples were incubated for 1 h at 37 °C on the plate by shaking (300 rpm), detected by HRP-coupled goat anti-human IgG Fc (A0170, Sigma-Aldrich, Darmstadt, Germany), and diluted 1:10,000 in Roti-Block™ blocking buffer (Carl Roth GmbH, Karlsruhe). The plate was incubated with tetramethylbenzidine (Carl Roth GmbH, Germany) for 5 min. The absorption at 450 nm was measured (Multiskan Go, Thermo Scientific, Germany | SkanIt Software, version 3.2) after terminating the reaction with 1 M hydrochloric acid. For the representation of glycan structures, the symbol nomenclature of the Consortium for Functional Glycomics was used: green circle, mannose; yellow circle, galactose; blue square, GlcNAc; yellow square, *N*-acetylgalactosamine (GalNAc); red triangle, fucose; purple diamond, Neu5Ac [41].

For the competitive ELISA, a 96-well plate was coated with 1 µg/mL BSA-α-gal conjugate (Dextra Laboratories Ltd., Reading, UK) in 100 mM ammonium carbonate, pH 9.6, overnight at 4 °C. After washing, the plate was blocked with Roti-Block™ blocking buffer (Carl Roth GmbH, Karlsruhe) for 1 h. The purified anti-α-Gal antibodies were preincubated in different excesses (0-, 5-, 10-, 20-, 50-, 100-fold molar excess) of biotinylated α-gal epitope and Galα1,3-gal for 1 h in PBS. After 1 h, the reaction vial was filled to 100 µL with deionized water. The mixtures of preincubated anti-α-Gal antibodies were added to immobilized BSAα-Gal and incubated for 1 h at 37 °C by shaking (300 rpm). For detection, the samples were incubated with HRP-coupled goat anti-human IgG Fc (Sigma-Aldrich, Darmstadt, Germany, A0170), diluted 1:10,000 in Roti-Block™ blocking buffer (Carl Roth GmbH, Karlsruhe). The plate was developed under the same conditions as the ELISA with biotinylated glycan epitopes.

#### 2.3.2. Erythrocyte Agglutination Assay

A round-bottom well plate was blocked with 1% BSA (*w/v*) in deionized water for 2 h at room temperature. Aliquots of 50 µL of samples with different anti-α-Gal antibody concentrations (100 µg/mL, 50 µg/mL, 25 µg/mL, 12.5 µg/mL, 6.25 µg/mL, 3.12 µg/mL, 1.56 µg/mL, 0.78 µg/mL) in physiological sodium chloride solution were

added, together with 50 μL of 0.5% (*v/v*) rabbit red blood cells (Dunn Labortechnik GmbH, Asbach, Germany) or human red blood cells (Deutsches Rotes Kreuz, Berlin, Germany) prepared by Fikoll (GE Healthcare, Braunschweig, Germany) density gradient centrifugation. Dilutions of erythrocyte suspensions were both made in physiological sodium chloride solution. Human erythrocytes were additionally incubated with blood group A antigen antibody (Thermo Fisher Scientific, Schwerte, Germany) in concentrations equal to anti-α-Gal antibodies in physiological sodium chloride solution. The well plate was incubated overnight at room temperature without shaking and was covered with a lid. Agglutination results were documented with the G-Box imaging system (Syngene, Cambridge, UK) and visually inspected.

### 2.3.3. Surface Plasmon Resonance (SPR)

SPR experiments were performed on a Biacore T200 (GE Healthcare, Freiburg, Germany). Biotinylated glycan epitopes were diluted in 10 mM hydroxyethyl–piperazine–ethane sulfonic acid (HEPES)/150 mM NaCl/0.02% (*v/v*) Tween 20, pH 7.5, and covalently coupled with streptavidin precoated SPR chips (Sensor Chip SA, BR100032) with immobilization levels of 200 response units (RU). Biotin was immobilized up to a response of 200 RU on one flow chamber as a reference. Purified anti-α-Gal antibodies were applied in different concentrations (157 nM, 52 nM, 17 nM, 6 nM, 2 nM, 0.67 nM) in 10 mM sodium acetate pH 4.5. Equal concentrations of human anti-factor VIII antibody (Coachrom Diagnostica, Maria Enzersdorf, Austria, MAB-HF8) were used as a negative control. A multicycle kinetic analysis was performed at 20 °C in 10 mM HEPES/150 mM NaCl/0.02% (*v/v*) Tween 20, pH 7.5, at a flow rate of 10 μL/min. The association and dissociation phases were monitored for 360 s, and the flow chamber surfaces were regenerated with two subsequent injections of 10 mM glycine, pH 2.0, for 10 s at 10 μL/min. The resulting binding data were fitted to a Langmuir 1:1 binding model by global fit analysis, which allowed the calculation of the dissociation constant $K_D$. First-order kinetics were assumed. The experiments were evaluated with the Biacore T200 evaluation software (version 3.1).

### 2.3.4. Western Blot

Gel electrophoresis was performed with tris-glycine gradient gels (8–16%, Anamed, Groß-Bieberau). Samples were pretreated with galactosidase or IdeZ. For galactosidase digest, 100 μg of glycoprotein, rebuffered to deionized water, were mixed with 5 mU of α-1,3,4,6-galactosidase (Prozyme, Ballerup, Denmark, GKX5007) and adjusted to 100 mM sodium citrate/phosphate, pH 6.0 with a total volume of 20 μL. The digestion was incubated for 18 h at 37 °C. For IdeZ digest, an amount of 50 μg protein was adjusted to 50 mM sodium phosphate, pH 7.5. The digestion was incubated at 37 °C for 30 min in a total reaction volume of 25 μL, including 1 μL of IgG-specific IdeZ protease (New England Biolabs, Ipswich, UK).

After enzymatical pretreatment the samples were adjusted to 0.72% Tris-HCl pH 6.8 (*w/v*), 2.5% sodium dodecyl sulfate (*w/v*), 10% glycerin (*v/v*), 10 mM dithiothreitol, 0.05% bromophenol (*w/v*) and incubated for 5 min at 95 °C. The dual color protein standard marker (Bio-Rad, Germany) was used. Running conditions were set to 100 V for 10 min, followed by 150 V for 50 min. After completed electrophoresis, the gel was blotted to a nitrocellulose membrane, as described by Towbin et al. [42]. A constant current of 250 mA was set for 1 h. Subsequently, the membrane was blocked overnight with 10% (*v/v*) RotiBlock™ blocking buffer (Carl Roth GmbH, Karlsruhe, Germany). The membrane was incubated with HRP-conjugated anti-α-Gal antibodies (1:4000) for 1 h at room temperature by soft shaking and washed 3 times with 10 mM Tris-HCl/0.1% tween 20 (*v/v*). The signal was detected via SuperSignal West Pico Chemiluminescent Substrate (Thermo Fisher Scientific, Schwerte, Germany) and documented by the G-Box imaging system (Syngene, Cambridge, UK).

### 2.3.5. Determination of IgG Subclasses

IgG subclasses of purified anti-$\alpha$-Gal antibodies were determined with the human IgG subclass profile kit (Thermo Fisher Scientific, Schwerte, Germany). The assay was performed according to the manufacturers' instructions. In brief, monoclonal antibodies, specific for one of the IgG subclasses 1, 2, 3, and 4 were preincubated with 2 µg/mL of anti-Gal antibody in dilution buffer for 5 min. A flat-bottom 96-well plate, precoated with anti-IgG antibodies, was loaded with the pre-incubated monoclonal antibodies. After 1 h incubation at room temperature by shaking at 300 rpm, followed by 3 washing steps, bound antibodies were detected by HRP-coupled anti-human IgG antibody (1:1000 in dilution buffer). Concentrations of IgG subclasses were calculated using four-parameter logistic regression (software: Excel, Microsoft Office, Version: 2010). IgG subclasses were additionally determined by radial immunodiffusion plates, which contained anti-IgG subclass antibodies (The Binding Site, Schwetzingen, Germany). Different concentrations of calibrators (IgG1: 140, 350, 840, 1400 µg/mL; IgG2: 80, 200, 480, 800 µg/mL; IgG3: 120, 300, 720, 1200 µg/mL; IgG4: 50, 125, 300, 500 µg/mL) were applied to the plates and the relative abundances of IgG subclasses were quantified relatively by measuring the diameter of visible precipitation rings as previously described by Dunn et al. [43].

### 2.3.6. Two-Dimensional (2D) Gel Electrophoresis

Purified anti-$\alpha$-Gal antibodies were diluted to a concentration of 300 µg/mL with re-hydration buffer (2.4 M thiourea, 8.4 M urea, 4.8% cholamidopropyl–dimethylammonium–propane sulfonate (Chaps) (*w/v*), 2.4% immobiline buffer (*v/v*), 0.36% dithiothreitol (*v/v*) and applied to lanes of a reswell tray. Immobilized pH gradient (IPG) gel strips (Immobiline$^{TM}$ DryStrip pH 6–11; GE Healthcare, Freiburg, Germany) were set on the samples in the lanes of the reswell tray and incubated overnight. The strips were removed from the box and placed on a DryStrip aligner. Both ends of the used strips were covered with moist paper. The loaded aligner was inserted into a focusing chamber (Multiphor II, GE Healthcare, Freiburg, Germany) and coated with mineral oil. Electrodes were put on, and the focusing process was started at a constant temperature of 20 °C, constant current (2 mA), constant power (5 W) with the following voltage gradient: 1 min 200 V, 15 min 200 V, 1 min 500 V, 15 min 500 V, 1 min 2000 V, 60 min 2000 V, 60 min 3500 V, 150 min 3500 V. Focused gel strips were incubated with electrophoresis equilibration buffer (6 M urea, 30% glycerin (*v/v*), 3% sodium dodecyl sulfate (*w/v*), 0.05 M Tris-HCl, 5 mM dithiothreitol) for 15 min. The solution was removed and replaced by equilibration buffer with 5 mM iodoacetamide and incubated for a further 15 min. Strips were stored in running buffer (250 mM Tris, 1.9 M Glycin, 35 mM SDS) before the second dimension was developed. The strips were placed on a tris-glycine gradient gel (8–16%, Anamed, Groß-Bieberau, Germany) and coated with warm 1% agarose (*w/v*). The dual color protein standard marker (Bio-Rad, Germany) was additionally applied. Running conditions were set to 100 V for 15 min, followed by 150 V for 75 min. After complete electrophoresis, the gel was stained with silver [44] and documented by the G-Box imaging system (Syngene, Cambridge, UK).

### 2.3.7. *N*-Glycan Analysis

Purified anti-$\alpha$-Gal antibodies (15 µg) were incubated in 1% (*v/v*) Rapigest solution (Waters GmbH, Eschborn, Germany) with 10% (*v/v*) tris(2-carboxyethyl)phosphine (Sigma Aldrich GmbH, Darmstadt, Germany) in deionized water and incubated for 5 min at 95 °C with shaking at 300 rpm. After cooling to room temperature, Rapid PNGase F (Waters GmbH, Eschborn Germany) was added to a total volume of 30 µL. The mixture was incubated for 30 min at 50 °C. A solution containing 12 µL of 9 mg of Rapifluor–MS dissolved in dimethylformamide was added and incubated for 5 min in the dark. The solution was diluted with acetonitrile to a final volume of 370 µL. The cleaning of labeled *N*-glycans was performed with the GlycoWorks HILIC µElution Plate (Waters GmbH, Eschborn, Germany). The cleaning procedure was performed according to the instructions delivered by the manufacturer. *N*-glycans were lyophilized in a vacuum centrifuge and resuspended in 10 µL of

a solution containing 94% (*v*/*v*), acetonitrile 3% (*v*/*v*) dimethylformamide and 3% water (*v*/*v*). Aliquots of 4 μL of *N*-glycan solution were analyzed by liquid chromatography coupled to mass spectrometry (LC-MS, Xevo®G2-XS QTof with AcquityH UPLC®Class, Waters GmbH, Eschborn, Germany) using an electrospray ionization source (High-Performance Zspray™-Multi mode source) in positive mode. LC separation was performed on a Waters Acquity UPLC Glycan BEH Amide column (130 Å, 1.7 μm, 2.1 × 150 mm), the temperature was kept constant at 60 °C, and a 35 min gradient of 25% A (50 mM ammonium formate pH 4.4)/75% B (acetonitrile) to 46% A/54% B was run. The mass spectrometer's instrument settings were adjusted for maximum sensitivity and detection selectivity (2750 V capillary voltage, 80 V cone voltage, 120 °C source temperature, 500 °C desolvation temperature, 50 L/h cone gas flow, 800 L/h desolvation gas flow). Calibration was performed with Flu Fibrinopeptide B, and a mass range between 750 Da and 2500 Da was recorded. The assignment of glycan structures was performed according to the respective retention times of the LC elution profile (GU units of glycan standards) and mass-to-charge ratios. For the representation of glycan structures, the symbol nomenclature of the Consortium for Functional Glycomics was used: green circle, mannose; yellow circle, galactose; blue square, GlcNAc; yellow square, *N*-acetylgalactosamine (GalNAc); red triangle, fucose; purple diamond, Neu5Ac [41].

## 3. Results and Discussion

### 3.1. Purification of Anti-α-Gal Antibodies

Anti-α-Gal antibodies were purified from a commercial human plasma-derived IgG concentrate (Octagam®, 100 mg/mL) by affinity chromatography on GlcNAcβ1,4-Galα1,3-Gal equipped sepharose. After washing with phosphate-buffered saline (PBS), bound anti-α-Gal antibodies were competitively eluted by 500 mM Gal. Subsequently, the column was washed with 100 mM glycine, pH 2.0 (Figure 1).

A threefold application of IgG was necessary to detect significant signals during elution. Collected fractions of several runs were pooled and the buffer was exchanged to PBS. The antibody concentration was adjusted to 1 mg/mL for further analysis. In earlier studies, anti-α-Gal antibodies were purified from human plasma by affinity chromatography using melibiose as affinity ligand [10,11,13]. Other affinity ligands such as α-Gal conjugated beads or bovine thyroglobulin were used [45,46].

The elution and wash fraction each revealed a relative protein amount of about 0.15% of totally applied IgG. This amount represents less than the content of about 1% anti-α-Gal antibodies in human IgG serum reported in the literature [47,48]. The lesser amount of anti-α-Gal antibodies in human serum may indicate only partial binding of α-Gal antibodies to the affinity column or an incomplete elution. Additionally, the high specificity of the affinity matrix may have led to a lower yield of anti-α-Gal antibodies because unspecific antibodies did not have sufficient binding affinity and eluted in the wash fraction. Furthermore, shape heterogeneity of the elution peak was observed, emphasizing the existence of different subclasses of anti-Gal antibodies.

The composition of purified anti-α-Gal antibodies was evaluated by sodium dodecyl sulfate–polyacrylamide gel–electrophoresis (SDS–PAGE) under reducing conditions (Figure 2).

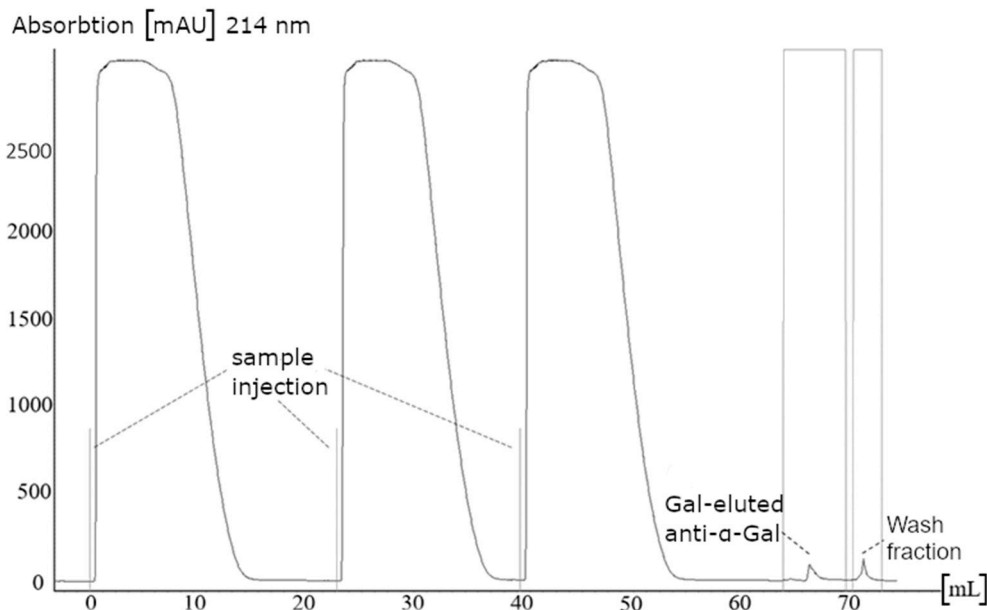

**Figure 1.** Exemplary chromatogram of the affinity purification of anti-Gal antibodies. Three samples of human IgG concentrate were applied to the affinity matrix (Galα1,3-Galβ1,4-GlcNAc-exposed sepharose). The absorption was monitored at 214 nm. Elution of bound antibodies was carried out by 500 mM Gal.

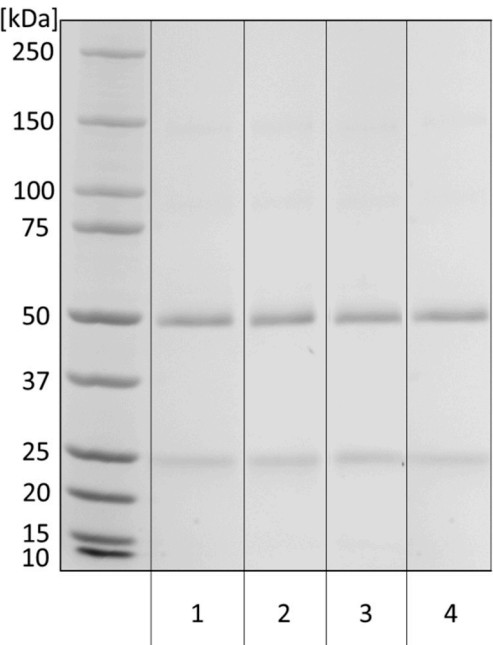

**Figure 2.** SDS–PAGE analysis of eluted anti-α-Gal antibodies to examine potential contamination with other proteins. A Tris-glycine gradient gel (8–16%) was loaded with Octagam® (1), unbound antibody (2), Gal-eluted antibody (3), and wash fraction (4) in reducing sample buffer. The gel was stained with Coomassie brilliant blue. The loaded amount of each sample was 2 μg.

All samples showed bands at 50 kDa (heavy chain of IgG) and 25 kDa (light chain of IgG) and a minor signal of 150 kDa (nonreduced IgG). No major non-IgG impurities were detected.

### 3.2. Anti-α-Gal Antibody Characteristics

3.2.1. Subclass Determination

The IgG subclass profile of anti-α-Gal antibodies was determined by ELISA utilizing four subclass-specific monoclonal antibodies. IgG2 was found to be the most abundant immunoglobulin isotype (83%), followed by IgG1 (14%) and tiny amounts of IgG3 (2%) and IgG4 (1%). In an orthogonal approach, radial immunodiffusion (RID) was used for the determination of IgG subclasses distribution. Anti-α-Gal sample and IgG subclass calibrators were applied to gel plates containing anti-subclass specific antibodies. Equal concentrations of the antibodies formed radial precipitation lines around the sample spots. Relative abundances of anti-α-Gal IgG subclasses were calculated by diameter comparison from samples and calibrators (Table 1). In this analysis, IgG2 is the most abundant isotype (76%), followed by IgG1 (24%), whereas IgG3 and IgG4 were not detected.

**Table 1.** Determination of IgG subclasses using RID plates. Measured diameters of calibrators and anti-α-Gal samples were squared and summarized with appropriately calculated concentrations. The relative proportions of IgG subclasses were determined by normalizing the calculated IgG concentrations. IgG3 and IgG4 were not detected (n.d.).

| Subclass | Calibrator Concentration [μg/mL] | Squared Diameter [cm$^2$] | | Sample Concentration [μg/mL] | Relative Part [%] |
|---|---|---|---|---|---|
| | | Calibrator | Sample | | |
| IgG1 | 1400 | 0.81 | 0.36 ± 0.06 | 492 ± 111 | 24 ± 6 |
| | 840 | 0.64 | | | |
| | 350 | 0.25 | | | |
| | 140 | 0.16 | | | |
| IgG2 | 800 | 0.64 | 0.55 ± 0.05 | 1422 ± 137 | 76 ± 7 |
| | 480 | 0.36 | | | |
| | 200 | 0.16 | | | |
| | 80 | 0.09 | | | |
| IgG3 | - | - | n.d. | - | - |
| IgG4 | - | - | n.d. | - | - |

The determined IgG subclass profile confirmed reports from the literature on the average amount of the subclasses of anti-α-Gal IgG (~10% for IgG1 and 80–90% for IgG2) in healthy individuals [49]. In contrast, the general subclass distribution of human IgG is about 60% for IgG1, 32% for IgG2, and 4% each for IgG3 and IgG4 [50], revealing a preference of anti-α-Gal antibodies for IgG2. Anti-α-Gal IgG2 is produced due to the natural stimulation of α-Gal-bearing bacteria in the intestinal flora [51]. This immune reaction is mediated by CD1d-receptor presenting lipid-linked carbohydrate antigens on APCs, which can be recognized by invariant natural killer T cells (iNKT) [52,53].

Anti-α-Gal IgG1 is produced after the occurrence of a tick bite and a subsequent exposure to α-Gal [54]. However, the anti-α-Gal IgG subclass distribution determined here was in accordance with the level of anti-α-Gal IgG subclasses of healthy people who did not suffer from tick bites [49]. The IgG subclass determination by RID was similar to the ELISA results. Nevertheless, in terms of quantitation, RID plates are reported to deliver partly inaccurate results, which may be the reason for minor differences between both assays [55]. The heterogeneity of the anti-α-Gal antibody sample was additionally shown by 2D electrophoresis (Figure 3).

Silver-stained spots at 50 kDa and 25 kDa are referred to as the heavy and the light chain of the reduced anti-α-Gal antibody. The spots represented the range of isoelectric points (IEPs) from pH 6 to pH 8 of anti-α-Gal antibodies, at least in parts due to the different subclasses or different subclass-specific glycosylation. The determined range of IEPs reflects the general range of IEPs for human IgG antibodies reported in the literature [56,57].

Galili et al. performed isoelectric focusing experiments and determined an IEP-range between pH 4 and pH 8.5 [1]. Our experiments revealed a more restricted range between pH 6 and pH 8, which underlined the possibility that anti-α-Gal subpopulations with different net charges exist.

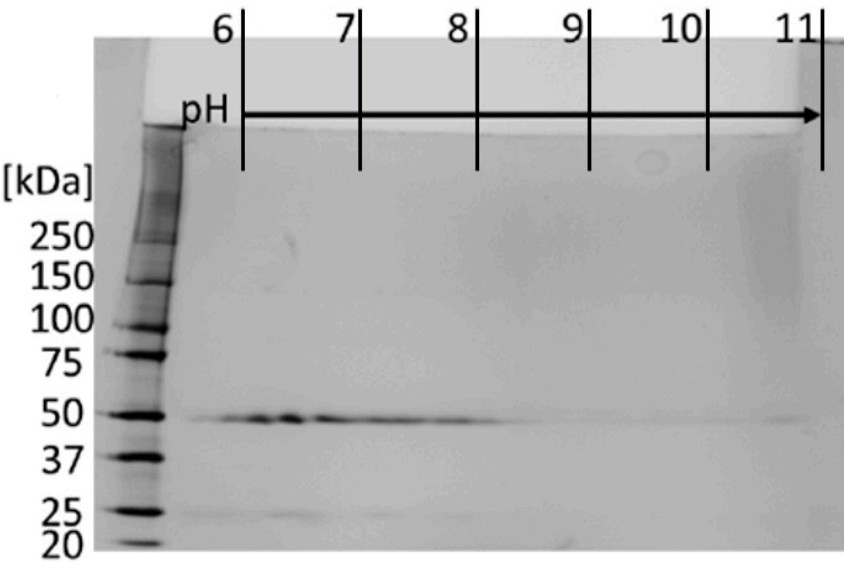

**Figure 3.** Two-dimensional gelelectrophoresis analysis of purified anti α Gal antibodies. Proteins were separated by isoelectric focusing, followed by SDS–PAGE (8–16% gradient gel) and silver staining.

3.2.2. Selectivity and Binding Affinities of Anti-α-Gal Antibodies

ELISA Assay

In order to evaluate the selectivity of purified anti-α-Gal antibodies, commercial biotinylated glycan epitopes were coated on a streptavidin precoated well plate, and the interaction between glycoconjugate and antibody was analyzed (Figure 4). Any glycan epitope without terminal α1,3-Gal did not reveal significant signals. Anti-α-Gal antibodies bound to the trisaccharide GlcNAcβ1,4-Galα1,3-Gal and, with a lower affinity, to the disaccharide Galα1,3-Gal. This result is in agreement with a study [58], in which was observed that the GlcNAc-residue downstream the Galα1,3-Gal disaccharide reinforced the binding via additional hydrogen bonds. Furthermore, binding to the structurally similar blood group B epitope was observed.

The purified anti-α-Gal antibodies are shown to be specific because epitopes without terminally α-Gal-linked Gal residues were not detected. A linkage variation evades antibody binding and indicates a very narrow binding pocket. The weak binding to the blood group B epitope further demonstrates that anti-blood group B antibodies are indeed a subpopulation of anti-α-Gal antibodies, as was already emphasized in the literature [59]. Individuals with blood type B produce lower titers of anti-α-Gal antibodies because of their self-tolerance to blood group B. Consequently, blood type B individuals have a higher susceptibility to α-Gal-bearing pathogens such as malaria [60]. Furthermore, blood group B individuals are less affected by red meat allergy [61]. To our current knowledge, it cannot be excluded that the occurrence of blood group B has an impact on tolerance of cetuximab, but it cannot be confirmed either.

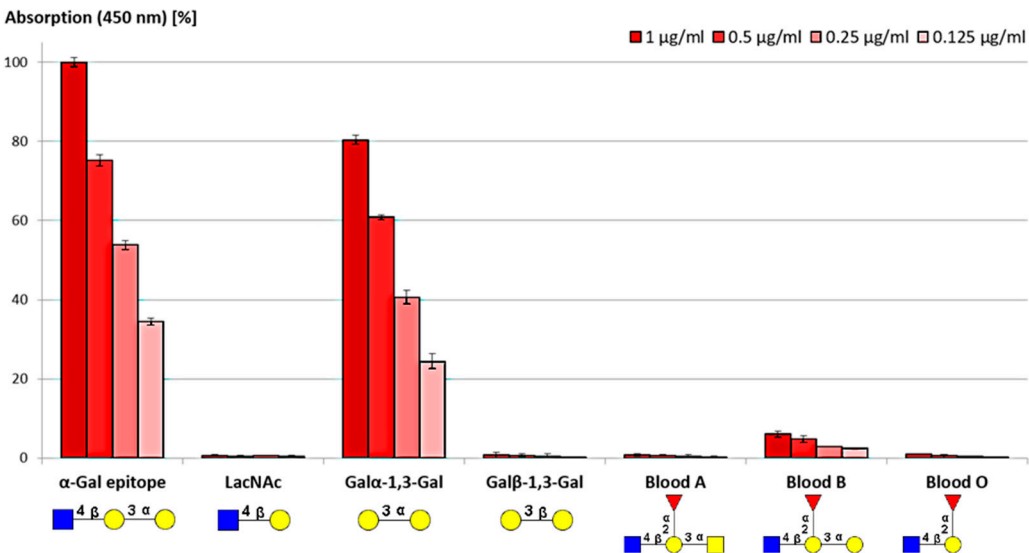

**Figure 4.** ELISA assay of anti-$\alpha$-Gal antibody binding to different commercial glycoconjugates serving as ligands in ELISA. Data of Gal-eluted anti-$\alpha$-Gal antibodies are shown as red bars. Each eluate was applied in the indicated four different concentrations. Data are presented as means $+/-$ SEM, n = 3.

Erythrocyte Agglutination

An erythrocyte agglutination assay was performed to provide further evidence of the selectivity of Gal-eluted anti-$\alpha$-Gal antibodies (Figure 5). Different concentrations of anti-$\alpha$-Gal antibodies were applied to a round-bottom well plate. Human and rabbit erythrocytes were added to the wells. Erythrocytes without antibodies served as a negative control. Cross-linked erythrocytes were visible as a fading surface in the round-bottom well when human blood group A erythrocytes were incubated with anti-human blood group A antibodies (Figure 5A). Unconnected cells slid down the round-bottom well, forming a dot (Figure 5, negative controls without the addition of antibodies). Anti-$\alpha$-Gal antibodies bound to rabbit red blood cells in a concentration-dependent manner (Figure 5C), whereas human erythrocytes were not bound by anti-$\alpha$-Gal antibodies at all (Figure 5B). These data confirm the literature, which emphasized that the $\alpha$-Gal epitope is only present on rabbit erythrocytes [62,63] but not on human erythrocytes. Furthermore, the data demonstrate that purified anti-$\alpha$-Gal antibodies did not show unspecific binding to the cell surface of human erythrocytes.

In former studies, the selectivity of purified anti-$\alpha$-Gal antibodies was shown by the agglutination of rabbit red blood cells or ELISA [1,64]. The binding of anti-$\alpha$-Gal antibodies to other glycans than $\alpha$-Gal cannot be completely ruled out due to the presentation of many other glycan epitopes, such as the AB0, Diego, or Kell antigens [65] on the surface of erythrocytes.

Binding Affinity

Surface plasmon resonance (SPR) measurements were performed to investigate dissociation constants of purified anti-$\alpha$-Gal antibodies. Biotinylated glycan epitopes were coated on a streptavidin precoated sensor chip. Measurements with blood group A and blood group O epitopes were omitted since no binding was detected via ELISA. LacNAc, Gal$\beta$1,3-Gal, and an anticoagulation factor VIII antibody were used as negative controls. Purified anti-$\alpha$-Gal antibodies only bound to the $\alpha$-Gal epitope ($K_D$ = 144 $\pm$ 20 nM, n = 3) and to the $\alpha$-Gal disaccharide ($K_D$ = 191 $\pm$ 18 nM, n = 3). The stabilizing effect of the additional GlcNAc-residue (GlcNAc$\beta$1,4-Gal$\alpha$1,3-Gal) did not significantly affect the interaction. However, a higher affinity of the trisaccharide, compared to the disaccharide (Gal$\alpha$1,3-Gal), was shown by an inhibition assay. Anti-$\alpha$-Gal antibodies were incubated with different excesses of either the trisaccharide or disaccharide. The binding to immobi-

lized bovine serum albumin (BSA)-α-Gal was determined (Figure 6). Twice the amount of disaccharide than trisaccharide was necessary for a 50% inhibition of BSA-α-Gal binding.

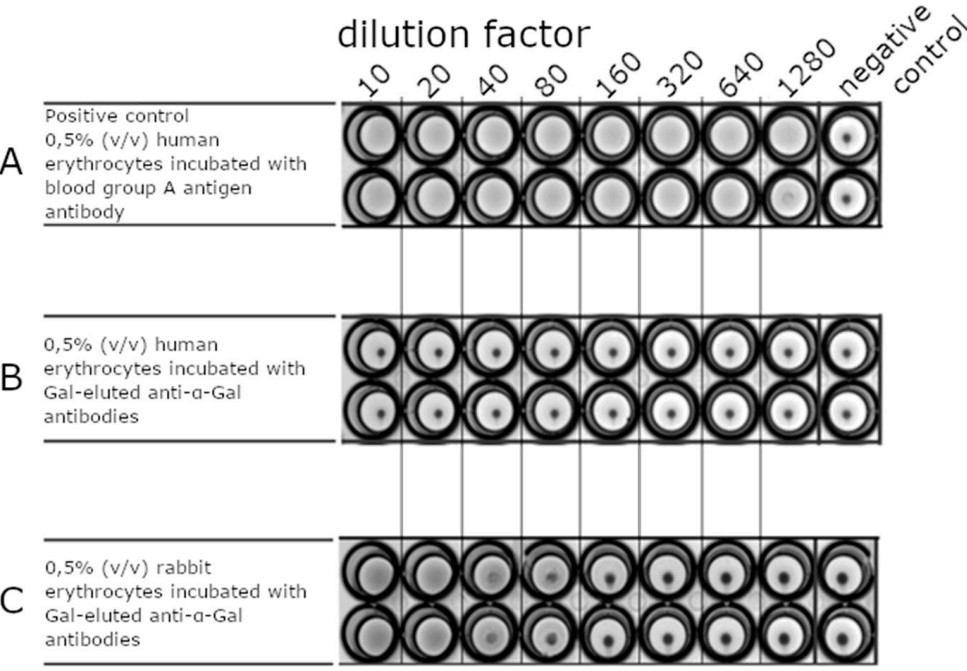

**Figure 5.** Agglutination of rabbit and human red blood cells to analyze the anti α Gal specificity toward the α Gal epitope. Agglutinations were carried out as duplicates. Different dilutions (10 to 1280) of antibody solution (1 mg/mL) were applied to a round-bottom well plate and incubated with human or rabbit erythrocytes. Agglutination takes place when a cross-linking between antibodies and erythrocytes becomes visible as a milky surface. The formation of a clot shows no agglutination. (**A**) Anti-blood group A antibody was applied on human red blood cells prepared from human plasma of an adult with blood group A as a positive control. (**B**) Anti-α-Gal antibody was applied on human red blood cells. The cells did not show any agglutination. (**C**) Anti-α-Gal antibody was applied on rabbit erythrocytes. The cells showed agglutination up to an antibody dilution factor of 80. Erythrocytes without additions were used as a negative control.

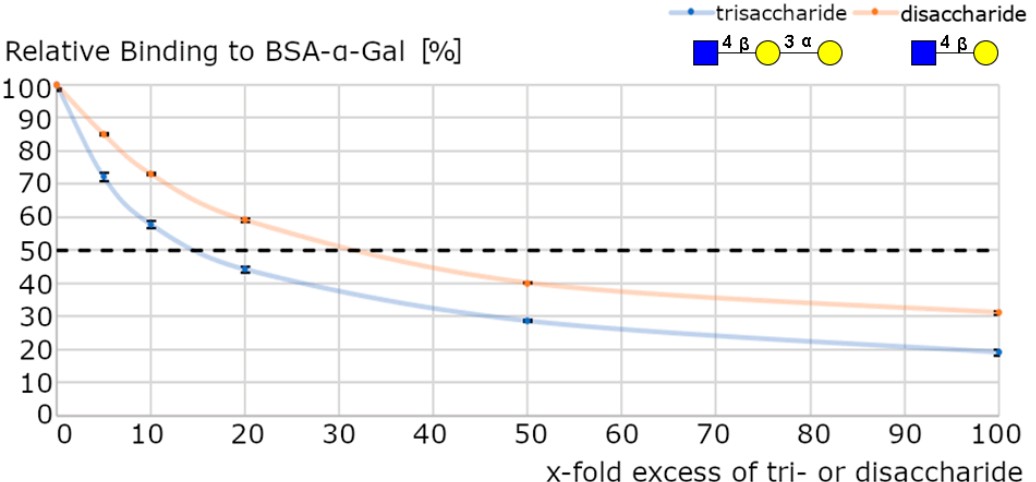

**Figure 6.** Anti-Gal antibody inhibition assay. A purified anti-Gal antibody was incubated with different excesses of trisaccharide (GlcNAc-β1,4-Gal-α1,3-Gal) or disaccharide (Gal-α1,3-Gal). Preincubated antibodies were subsequently incubated with immobilized BSA-α-Gal in a well-plate. Data are presented as means +/− SEM, n = 3.

The affinity of anti-$\alpha$-Gal antibodies to $\alpha$-Gal is high, compared to the dissociation constant of other carbohydrate-specific antibodies. The dissociation constant $K_D$ of high-affinity monoclonal antibodies specific for chlamydial lipopolysaccharide was reported in the range of about 500 to 700 nM [66]. That was a 2.5-to-4-times higher dissociation constant than the $K_D$ that was calculated for the binding of anti-$\alpha$-Gal antibodies to $\alpha$-Gal in our assay. Compared to dissociation constants of specific therapeutical monoclonal antibodies, which were in the picomolar range for their corresponding antigen, the affinity of anti-$\alpha$-Gal was lower by a factor up to 2000 [67]. To include potential effects of polypeptide backbones on the binding affinity of anti-$\alpha$-Gal antibodies, we immobilized bovine thyroglobulin on a carboxylmethyl (CM) 5 sensor chip and measured the binding affinity to purified anti-$\alpha$-Gal antibodies. The dissociation constant was determined as 1.6 nM, which suggests a strong positive influence of the protein part on anti-$\alpha$-Gal antibody affinity. Binding studies of anti-$\alpha$-Gal IgE, bovine thyroglobulin, and human serum albumin coated with $\alpha$-Gal determined dissociation constants of 36 nM (bovine thyroglobulin) and 363 nM (albumin) [64]. Experiments to determine the binding affinity of an engineered antibody against *N*-glycolylneuraminic acid (Neu5Gc) on proteins resulted in dissociation constants at about 1 $\mu$M [68]. This is a 1000-fold higher dissociation constant than the $K_D$ values determined here ($K_D$ between anti-Gal and bovine thyroglobulin = 1.6 nM), indicating a very high affinity of purified anti-$\alpha$-Gal antibodies toward $\alpha$-Gal epitopes. The high specificity of the purified antibodies is the decisive difference to the recombinant anti-Gal antibody variants M86 and G-13. In contrast to the simple enrichment and purification of human, highly specific anti-Gal antibodies, the expression of the recombinant variants leads to numerous problems such as autolysis or self-agglutination [59,69–71].

### 3.2.3. *N*-Glycosylation Profile of Anti-$\alpha$-Gal Antibodies

The glycosylation of antibodies has been extensively studied [72–74]. To characterize the *N*-glycosylation profile of purified anti-$\alpha$-Gal antibodies, their *N*-glycans were enzymatically released with PNGase F. Enzymatically released *N*-glycans were labeled with fluorescent RapiFluor MS reagent, separated via hydrophilic interaction liquid chromatography (HILIC), and applied to the mass spectrometric analysis. The resulting *N*-glycan profile of purified anti-$\alpha$-Gal antibodies revealed a typical human IgG-like glycosylation (Figure 7).

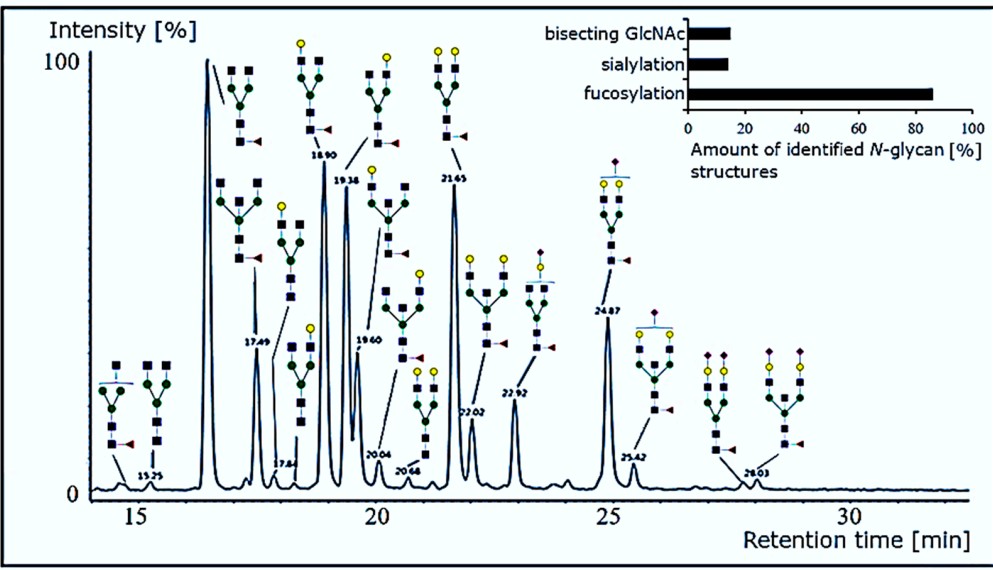

**Figure 7.** *N*-glycan profile of anti-$\alpha$-Gal antibodies. PNGase F-released *N-glycan*s were separated via hydrophilic interaction liquid chromatography. Peak identification was achieved by independent Q-TOF mass spectrometry. A schematic *N-glycan* representation is given for signals with a relative peak area >2% only. Evaluations were carried out using the Waters UNIFI 1.9.2 software.

Out of the detected *N*-glycans, 86% were fucosylated. The most prominent structure was the complex biantennary, fucosylated glycan without antennary galactoses (G0) followed by the mono- and fully galactosylated forms (G1 and G2). Bisecting GlcNAc (15% of all structures) and sialylated glycans (12% partial, 1% full) represent further elements. To our knowledge, the *N*-glycosylation profile of anti-α-Gal antibodies was not reported before. Especially the high grade of core fucosylation (86%), the low grade of sialylation (13%), and a moderate grade of bisection (15%) are typical for human IgG [75,76]. Glycan structure–function relationships, particularly for antibodies, are a major issue to date [77]. A fully elucidated *N*-glycan profile of anti-α-Gal may help prospective therapeutic applications of the antibody or optimize its recombinant expression.

### 3.3. Verification of the Anti-α-Gal Suitability as Detection Antibody

In this study, purified anti-α-Gal antibodies were investigated for their applicability in Western blot analyses applying HRP-induced chemiluminescence for detection. For proof of concept, the α-Gal-carrying glycoproteins bovine thyroglobulin, cetuximab, and a synthetically produced BSA-α-Gal-conjugate were analyzed utilizing the purified anti-α-Gal fraction as primary antibodies (Figure 8A).

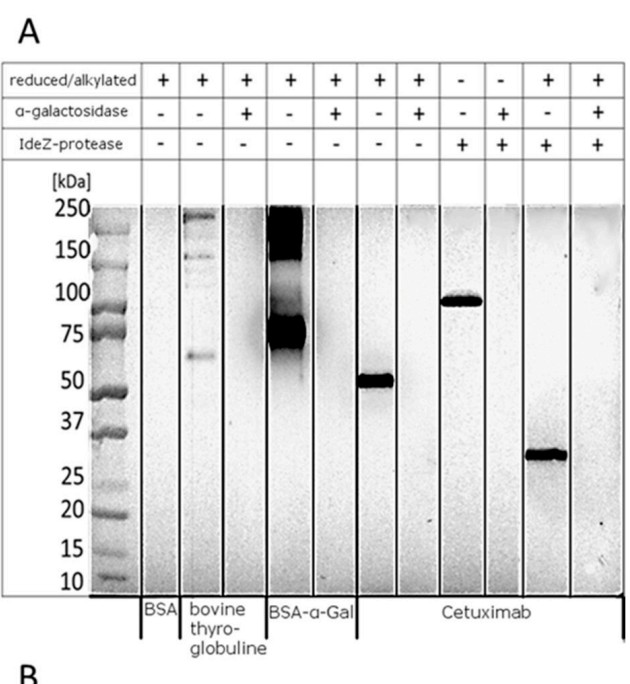

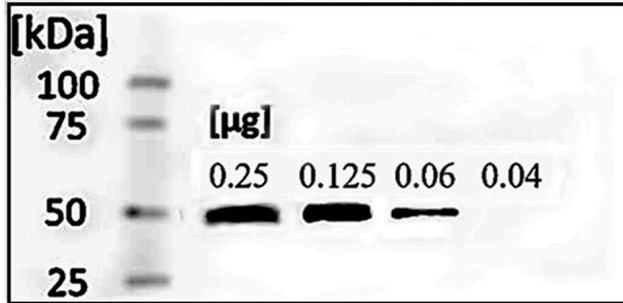

**Figure 8.** (**A**) Western blot analysis of glycoproteins, developed by HRP-coupled anti-α-Gal antibodies. BSA (negative control), bovine thyroglobulin, BSA-α-Gal, and cetuximab (1 μg each) were separated on an 8–16% gradient gel and blotted. Previous digestions by α1,3 galactosidase and IdeZ-protease are indicated (- undigested, + digested); (**B**) Western blot analysis of cetuximab developed by anti-α Gal-HRP conjugate (1:4000).

After reduction and alkylation, bovine thyroglobulin showed bands from 75 to over 250 kDa, representing products caused by the reduction of disulfide bridges, which is in line with reports from the literature [78]. Synthetic BSA-α-Gal revealed a monomer at about 66 kDa and bands from 150 kDa to over 250 kDa, most likely due to aggregation. Cetuximab showed a band at 50 kDa, indicating glycosylation at the heavy chain. After cleavage at the hinge region, catalyzed by IdeZ protease, the Fab fragment was detected at 100 kDa. After reduction, this signal was shifted to ~25 kDa. Anti-α-Gal antibodies did neither bind to any α1,3-galactosidase-treated proteins nor the negative control (BSA).

Different amounts of cetuximab were separated by SDS–PAGE, blotted, and developed with HRP-conjugated anti-α-Gal antibodies (Figure 8B) to evaluate the detection limit of anti-α-Gal antibodies. Cetuximab was detectable down to a value of 0.04 µg (0.28 pmol). The known molar amount of four α-Gal epitopes for cetuximab [79] reveals a detection limit of 1.12 pmol of α-Gal epitopes.

Detailed glycan analysis requires a lot of time, expensive equipment, and skilled scientists. This assay is fast and comparatively uncomplicated. However, an obvious disadvantage of the Western blot assay is that only glycoproteins of sufficiently high mass can be analyzed due to the previous separation via gel electrophoresis. Glycopeptides are too small to be detected with Western blot, whereas mass spectrometric detection of glycopeptides or even smaller molecules has been carried out a lot of times [80–82]. Therefore, selected model glycoproteins with high amounts of α-Gal epitopes were analyzed via Western blot in this study only. Purified anti-α-Gal antibodies can also be used for quantification of the amount of α-Gal in specific glycoproteins in ELISA assays with higher sensitivity (see Figure 4 for use of α-Gal antibodies in ELISA in general). For this, more therapeutic glycoproteins with low α-Gal content should be analyzed to identify the detection limit of the anti-α-Gal antibody.

For example, the monoclonal antibodies palivizumab, dinutuximab, necitumumab, and elotuzumab, which are produced in α-Gal synthesizing murine cells [83] may require continuous monitoring of the α-Gal content during bioprocesses.

**Author Contributions:** Writing—original draft preparation, visualization, investigation, conceptualization, methodology A.Z.; conceptualization, methodology, validation, investigation, data curation, writing—review and editing, J.R.; conceptualization, methodology, validation, investigation, data curation, writing—review and editing, G.K.; supervision, project administration, writing—review and editing, S.H.; supervision, project administration, writing—review and editing, M.K.P. All authors have read and agreed to the published version of the manuscript.

**Funding:** This research received no external funding.

**Institutional Review Board Statement:** Not applicable.

**Informed Consent Statement:** Not applicable.

**Conflicts of Interest:** The authors declare no conflict of interest.

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
