# Peer review of "Purification and Characterization of Antibodies Directed against the α-Gal Epitope"

_2673-6411, doi:10.3390/biochem1020008_

Round 1

Reviewer 1 Report

This is an excellent piece of methodological work that certainly deserves publication, since valuable procedures to isolate and characterize specific antibodies from human samples are highly required. Moreover, indeed the alpha-Gal epitope is particularly relevant, but elusive as well.

I think that the manuscript can be published with just a minor clarification, concerning the definition of the sample used for the isolation procedure. On page 3, line 98 the authors state that “Octagam sample was applied [to the affinity column] 3 times in a row”. At this level, it is unclear if the used 3 samples of 5 mL (i.e., ONE sample of 15 mL) or maybe one single 5-mL sample was recycled three times onto the column.

A clarification is obtained in the legend of Fig. 1, where it is much clearer that “Three samples of human IgG concentrate were applied to the affinity matrix”. The reason is finally clarified few lines below (page 8, line 306): “A threefold application of IgG was necessary to detect significant signals during elution”. 

This minor aspect should be fixed, I guess that the volume of a commercial sample of Octagam is 5 mL. Moreover, it would be useful to specify the total IgG content of this commercial product. This piece of information will help in evaluating the overall results, including the (low) yield (e.g., line 312: “The elution and wash fraction each revealed a relative protein amount of about 0.15% of 312 totally applied IgG”.)

Author Response

This is an excellent piece of methodological work that certainly deserves publication, since valuable procedures to isolate and characterize specific antibodies from human samples are highly required. Moreover, indeed the alpha-Gal epitope is particularly relevant, but elusive as well.

The authors thank the reviewer for these very positive comment to our work.

I think that the manuscript can be published with just a minor clarification, concerning the definition of the sample used for the isolation procedure. On page 3, line 98 the authors state that “Octagam sample was applied [to the affinity column] 3 times in a row”. At this level, it is unclear if the used 3 samples of 5 mL (i.e., ONE sample of 15 mL) or maybe one single 5-mL sample was recycled three times onto the column.

A clarification is obtained in the legend of Fig. 1, where it is much clearer that “Three samples of human IgG concentrate were applied to the affinity matrix”. The reason is finally clarified few lines below (page 8, line 306): “A threefold application of IgG was necessary to detect significant signals during elution”. 

We have clarified this point now in line 98-101. Three samples of 5 ml each were applied to the affinity column.

This minor aspect should be fixed, I guess that the volume of a commercial sample of Octagam is 5 mL. Moreover, it would be useful to specify the total IgG content of this commercial product. This piece of information will help in evaluating the overall results, including the (low) yield (e.g., line 312: “The elution and wash fraction each revealed a relative protein amount of about 0.15% of 312 totally applied IgG”.)

The stated purity of Octagam is 95% (see line 88). Total content of human IgG in Octagam could be calculated from this data.

Reviewer 2 Report

The paper entitled “Purification and characterization of antibodies directed against the α-Gal epitope” is a very nice piece of work, and one that surely merits being published. Well-written, data clearly presented and easy to understand. I have only minimal remarks.

My main comment is just a general crusade in the antibody field, and of course it’s not something to be done for this paper in particular. But it would be wonderful to try to get these antibodies sequenced, and make a pure multiclonal antibody to use as detection tool. I know that technically this can be very difficult, but it’d be great to see. I know on page 15 you comment anti-Gal rAbs are inferior, but you didn’t comment on the reason why this could be the case (I’m not asking to such comment to be included on the paper; it’s just a personal doubt).

My few comments are below:

  • Pg 3, line 89: “intravenous human IgG concentrate”. It’s useful to put here the original antibody concentration of Octagam as well. I was puzzled, when I first read, about the antibody concentrations (they were a bit too high, given what one can get commercially with research antibodies. And then I saw Octagam comes as 100 mg/ml solution on page 8);
  • There are a few places in which the decimal has a comma, and not a dot (pg 5, line 207; Table 1; pg 15, line 482);
  • Figure 2: Pg 9, line 327, missing a ‘-’ on ‘8 16%’. Still on Figure 2: any reason why all fractions have absolutely same intensities?
  • On the last paragraph, you comment on the validity of the antibodies as detection tool by WB, and the potential problems. Would it be possible to apply them in a simple ELISA assay instead? Perhaps it’s a naive question, but ELISA is a bit superior as detection tool, even more when going to different concentrations, multiple targets and small molecules. It’d be nice to have 1-2 sentences on this (if the point is valid, of course. Otherwise, just ignore the comment).

Author Response

The paper entitled “Purification and characterization of antibodies directed against the α-Gal epitope” is a very nice piece of work, and one that surely merits being published. Well-written, data clearly presented and easy to understand. I have only minimal remarks.

The authors appreciate the very positive feedback of the reviewer.

My main comment is just a general crusade in the antibody field, and of course it’s not something to be done for this paper in particular. But it would be wonderful to try to get these antibodies sequenced, and make a pure multiclonal antibody to use as detection tool. I know that technically this can be very difficult, but it’d be great to see. I know on page 15 you comment anti-Gal rAbs are inferior, but you didn’t comment on the reason why this could be the case (I’m not asking to such comment to be included on the paper; it’s just a personal doubt).

The authors completely agree that one of the next steps of our project should be sequencing of the purified a-Gal antibodies and generation of new recombinant a-Gal antibodies with hopefully higher affinity. The commercially avaliable a-Gal antibodies derive from animal immunization, which seems to be less effective than “immunization” of a-Gal-free human beings.

My few comments are below:

Pg 3, line 89: “intravenous human IgG concentrate”. It’s useful to put here the original antibody concentration of Octagam as well. I was puzzled, when I first read, about the antibody concentrations (they were a bit too high, given what one can get commercially with research antibodies. And then I saw Octagam comes as 100 mg/ml solution on page 8);

The concentration of Octagam has been inserted at line 89. 100 mg/mL indeed is a much higher concentration compared to research antibodies, but it is necessary for successful IVIG therapy.

There are a few places in which the decimal has a comma, and not a dot (pg 5, line 207; Table 1; pg 15, line 482)

The decimals in Table 1 were replaced by commas.

Figure 2: Pg 9, line 327, missing a ‘-’ on ‘8 16%’. Still on Figure 2: any reason why all fractions have absolutely same intensities?

The hyphen was added. Furthermore, the legend of figure 2 was complemented by the statement “The loaded amount of each sample was 2 µg.” to explain equal intensities.

On the last paragraph, you comment on the validity of the antibodies as detection tool by WB, and the potential problems. Would it be possible to apply them in a simple ELISA assay instead? Perhaps it’s a naive question, but ELISA is a bit superior as detection tool, even more when going to different concentrations, multiple targets and small molecules. It’d be nice to have 1-2 sentences on this (if the point is valid, of course. Otherwise, just ignore the comment).

The authors agree that it is worth to mention the additional opportunity to apply a-Gal antibodies in ELISA. We have therefore added some statements about this in lines 556-562.